# Photon Dissipation as the Origin of Information Encoding in RNA and DNA

**DOI:** 10.3390/e22090940

**Published:** 2020-08-27

**Authors:** Julián Mejía Morales, Karo Michaelian

**Affiliations:** 1Postgrado in Physical Sciences, Instituto de Física, Universidad Nacional Autónoma de México, Cto. de la Investigación Científica, Cuidad Universitaria, Mexico City C.P. 04510, Mexico; julianmejia@ciencias.unam.mx; 2Department of Nuclear Physics and Application of Radiation, Instituto de Física, Universidad Nacional Autónoma de México, Cto. de la Investigación Científica, Cuidad Universitaria, Mexico City C.P. 04510, Mexico

**Keywords:** entropy, entropy production, non-equilibrium thermodynamics, information encoding, nucleic acids, DNA, RNA, origin of life, origin of codons, amino acids, stereochemical era, photon potential

## Abstract

Ultraviolet light incident on organic material can initiate its spontaneous dissipative structuring into chromophores which can catalyze their own replication. This may have been the case for one of the most ancient of all chromophores dissipating the Archean UVC photon flux, the nucleic acids. Oligos of nucleic acids with affinity to particular amino acids which foment UVC photon dissipation would most efficiently catalyze their own reproduction and thus would have been selected through non-equilibrium thermodynamic imperatives which favor dissipation. Indeed, we show here that those amino acids with characteristics most relevant to fomenting UVC photon dissipation are precisely those with greatest chemical affinity to their codons or anticodons. This could provide a thermodynamic basis for the specificity in the amino acid-nucleic acid interaction and an explanation for the accumulation of information in nucleic acids since this information is relevant to the optimization of dissipation of the externally imposed thermodynamic potentials. The accumulation of information in this manner provides a link between evolution and entropy production.

## 1. Introduction

Explaining the origin of the genetic code is a difficult problem since this is ultimately related with deciphering the problem of the origin of life itself. From the rather incredulous perspective of life as a fortuitous enzyme-catalyzed auto-catalytic reproductive event, the problem arises in that sufficient information must have somehow been acquired in the incipient genome before a necessarily faithful replication event could have taken place. Due to the complexity of such an event, involving denaturing, chiral selection, and extension, it is highly unlikely that that the sufficient information could have been generated at random.

A number of theories have attempted to address this information problem by considering the origin of the association between amino acids and their cognate codons or anticodons. The “stereochemical theory” suggests that amino acids became linked with their codons or anticodons based on stereo-chemical affinity [1,2,3], while the “frozen accident theory” [4,5] suggests that such links arose purely by chance but once established would have remained, so since any changes would have been highly deleterious to protein construction and surely detrimental to the organism. The “co-evolutionary theory” [6] suggests instead that the structure of the codon system is primarily an imprint of the prebiotic pathways of amino acid construction, which remain recognizable in contemporary enzymatic pathways of amino acid biosynthesis.

Such theories only partially address the problem of the origin of information in the genetic code since, since although they provide plausible reasons for the association between particular amino acids and their cognate codons, they do not suggest a physical–chemical basis for the specificity of such an association and how this relates to the origin of life itself; i.e., to enzyme-less replication, proliferation and evolution. Carl Woese [1] recognized this early on and emphasized that this was (and still is) an unresolved problem; that of uncovering “*the basis of the specificity between amino acids and codons*” in the genetic code.

In this paper, by incorporating the stereochemical theory into the framework of non-equilibrium thermodynamic theory, we are able to provide a novel understanding of the physical–chemical basis of the specificity of these associations between codons/anticodons and amino acids. The basis we propose for this specificity is that the the fundamental molecules of life, and in particular the complexes of nucleic acid with amino acids, were synthesized through microscopic dissipative structuring under the Archean UVC solar photon spectrum prevailing at Earth’s surface during the Archean to dissipate this spectrum into heat, i.e., to produce entropy [7,8,9,10]. We show that those complexes most efficient at photon dissipation would most efficiently catalyze their own reproduction, leading to a dissipation–replication relation. There is a non-equilibrium thermodynamic imperative which favors the amplification of fluctuations which lead to stationary states (disipative structures) with greater dissipation efficacy [11,12,13,14,15,16,17]. Information related to which nucleic acid – amino acid complexes provided most efficient photon dissipation would thus gradually have begun to be incorporated into the primitive genetic code.

## 2. Foundations

Clasical Irreversible Thermodynamic (CIT) theory, formulated by Lars Onsager [11,12,18], Ilya Prigogine [19], and others indicates that all irrevesible processes arise, persist, and even evolve to dissipate a generalized thermodynamic potential, i.e., to produce entropy. Glansdorff and Prigogine have shown how a non-linear system over which a generalized thermodynamic potential is imposed can “self-organize” into structures (or more correctly, processes) which can break symmetry in both space and time if this increases dissipation [10,14]. This organization of material is known as “dissipative structuring” in non-equilibrium thermodynamic language. Biologists are acutely aware of a sub-set of this structuring and refer to it simply as“life”.

Macroscopic dissipative structuring leads to macroscopic processes such as hurricanes, winds, ocean currents and the water cylce. However, dissipative structuring can also occur over molecular internal degrees of freedom; electronic or vibrational coordinates, spin coordinates, and reaction coordinates (ionizations, deprotonations, charge transfer, disassociations, isomerizations, tautomerizations, rotations around covalent bonds, sigmatrophic shifts, etc.), exciplex and excimer formation, etc. In the case of structuring involving molecular configurational degrees of freedom, the macroscopic variables are the concentration profiles of the precursors, intermediates and product molecules [10].

The most important generalized thermodynamic potential (source of free energy) at the Earth’s surface today, and also at the origin of life at ∼3.85 Ga, is the solar photon potential; the sun’s spectrum contains a large amount of free energy with respect to the low free energy spectrum of the cosmic black-body radiation filling space surrounding Earth. During the late Hadean and early Archean, sunlight surpassed all other forms of free energy (hydrothermal vents, lightening, chemical potentials, shock waves, etc.) by at least three orders of magnitude [20]. The most important dissipative structures existing on Earth’s surface today are the organic pigments in water solvent [21] and these are responsible for approximately 63% of the total entropy production resulting from Earth’s interaction with its solar environment [22]. The coupling of the heat of dissipation of the solar photons in organic pigments to other macroscopic irreversible processes such as ocean and wind currents and the water cycle, accounts for the majority of the rest of the entropy production on Earth [23].

Without the pre-existence of complex biosynthetic pathways, the photon wavelengths that could have bootstrapped life through dissipation at life’s origin at the end of the Hadean or early Archean must have involved the long wavelength UVC and UVB regions, where there is enough free energy to make and break carbon covalent bonds, but not enough to fragment and thereby destroy organic molecules. A long wavelength UVC component (peaking at ∼260 nm) of the solar spectrum indeed penetrated to the surface of Earth during the later Hadean and throughout the Archean with an integrated intensity of up to 5 W/m${}^{2}$ midday at the equator [7,24]. Evolving life was exposed to this UVC photon potential for at least 1000 million years (1 Ga) until approximately 2.7 Ga when organisms performing oxygenic photosynthesis became abundant enough to overwhelm the natural abiotic oxygen sinks [25].

Corroborating evidence for the exposure of life to this UVC photon potential, and indeed life’s thermodynamic preoccupation with its dissipation, can be found in the fundamental molecules of life (those common to all three domains of life). Most of these absorb and dissipate this light into heat with great efficiency (see Figure 1) [21,26]. We have therefore suggested that the fundamental molecules originated in the later Hadean or early Archean as microscopic dissipative structures to perform this thermodynamic function [7,8,9,10,26].

In the photochemical non-linear reactor system we are considering here, consisting of the surface microlayer of the ocean or lake under a constant flux of UVC photons and a constant inflow of precursor molecules, and a constant outflow of heat and product molecules, we refer to the final product molecules (pigments) absorbing and dissipating the incident photon flux as “microscopic dissipative structures” since the structuring remains even after the removal of the impressed generalized thermodynamic potential (the photon flux) due to strong inter-atomic forces [9,10].

The proliferation of these pigments over the whole surface of Earth, a hallmark of biological evolution, can be explained through the auto-catalytic nature of these pigments in dissipating the same UVC potential that produced them photochemically [10,26,27]. Analogously to what Prigogine [28] has demonstrated using CIT theory for auto-catalytic chemical reactions, the concentration of a photochemical product (chromophore or pigment) will increase much beyond its expected equilibrium concentration if the product acts as a catalyst for the dissipation of the same impressed photon potential that produced it [27]. For fundamental chromophores of the Archean such as the nucleic acids absorbing in the UVC, this leads to a direct relation between dissipation and replication. Proliferation of these fundamental molecules in this manner over the whole of Earth’s surface is thereby driven by the thermodynamic imperative of increasing the global entropy production of Earth in its solar environment. Although dissipative structuring of material may lead to a reduction in its entropy, that of the system plus environment invariably increases, thereby respecting the second law of thermodynamics.

Since microscopic dissipative structuring can be persistent, i.e., structuring remains even after the removal of the impressed external potential, information concerning the impressed potential and the molecular structuring required for its dissipation becomes programmed into the structure of the material. The cumulative historical information regarding the external generalized chemical potentials, and the blueprint for construction of the biosynthetic pathways needed for the production of structures (e.g., chromophores and their supporting structures) required for the dissipation of these potentials, is today known as the genome. Such a microscopic mechanism with inherent persistence in structure, along with the production of variants through microscopic mutation of the genome, allows the system to “evolve”, i.e., to dynamically adapt the dissipative structure towards optimal dissipation, to track changes in the external potential, or to resume dissipation should the external potential return after temporal absence (for example, after the overnight extinction of the solar photon potential).

The mechanism (to be described below) relating replication with dissipation, under the non-equilibrium thermodynamic imperative of increasing the global entropy production, allows the system to evolve to ever greater efficacy in photon dissipation (by, for example, increasing the areal coverage of pigments over Earth’s surface, or increasing the wavelength region of dissipation) often with a corresponding increase in system complexity, eventually arriving at the complex global dissipative structure of today known as the *biosphere* which involves the coupling of biotic with abiotic dissipative processes [29]. Before detailing this mechanism relating replication with dissipation, we first discuss some of the salient properties of the fundamental molecules of life which provide evidence for, and are crucial to, the viability of such a mechanism.

## 3. Salient Properties of Amino Acid, Nucleic Acids, and Their Complexes

### 3.1. Absorption and Dissipation

The fundamental molecules of life (those molecules common to all three domains of life) and their associations in complexes are examples of microscopic self-organized dissipative structures which formed under the Archean UVC photon potential [7,8,9,10,26,27]. Evidence for this can be found in the strong absorption cross sections of the fundamental molecules, with absorption maxima lying exactly within the predicted UVC window of Earth’s atmosphere during the Archean [24] (Figure 1), and in the existence of inherent conical intersections giving these molecules an extraordinarily rapid non-radiative electronic excited state decay into heat [10,21,30].

The nucleobases, and also their polymerized agglomerations of single and double strand RNA and DNA, absorb and dissipate UVC light and can be formed from simpler precursor molecules such as hydrogen cyanide HCN in water by the very same UVC photons that they eventually dissipate with such efficacy [9,10,21]. An example of such microscopic dissipative structuring under UVC light is the generic photochemical pathway to the purine nucleobases first discovered by Ferris and Orgel in 1966 [31]. Although the particular photochemical reaction pathways to the multitude of other fundamental molecules (UVC pigments) still remain to be discovered, the existence of microscopic dissipative structuring is evidenced, for example, by the large quantity of organic pigment molecules found throughout the cosmos wherever UV light is available [32]. Today, a similar situation exists on Earth, with pigments dissipating in the near UV and visible which are constructed through more complex biosynthetic pathways but remain, nonetheless, microscopic dissipative structures dissipating the same visible light that provided the free energy for their production.

### 3.2. Amino Acid Affinity to Codons (Or Anti-Codons) of DNA and RNA

Stereochemical theories for the origin of the genetic code propose that chemical affinity between amino acids and nucleic acids was the historical basis for the present association between codons and/or anticodons and cognate amino acids [2,3,33,34,35,36,37,38,39,40,41,42,43,44,45,46]. These works present sufficient evidence to suggest that, at some early stage in the evolution of life, amino acids had chemical affinity to specific polynucleotides which later evolved into the genetic code.

Although evidence for the chemical association of the twenty common amino acids with RNA or DNA exists, not all associations are of the same character or strength. There are eight distinct types of associations between amino acids and nucleotides or their polymerizations into nuclei acids; (1) the π-cation bond [45,47], (2) π-stacking of aromatic structures [48], (3) Coulomb charge interaction [49], (4) hydrophobic aggregation [38], (5) Van der Waals interaction, (6) hydrogen bonding [50], (7) stereochemical (for example, substitution of nucleotides by amino acids of similar structure [40,51]), and (8) the possibility of amino acids synthesis from α-ketos acids bound to dinucleotides [52].

Of these eight forms of association, four are non-specific, meaning that the amino acids do not bind to any particular sequence along the single or double strand DNA or RNA. However, the other four types of associations; the π-cation bonding, hydrogen bonding, stereochemical affinity and amino acid synthesis from α-ketos, are specific, having high specificity, i.e., bind the amino acids almost unequivocally to either their cognate codons or anticodons.

One of the non-specific associations between nucleic acids and amino acids is due to the aliphatic nature of the side chain of the amino acids which can attach themselves, for example within the grooves of the nucleic acid, through van der Waals and hydrophobic interactions [3,44]. Another non-specific interaction is that between aromatic amino acids and nucleobases due to stacking interactions that occur when amino acid side chains contain charged sites like indole rings or aromatic rings, for example, Trp, Tyr, Phe and His [48]. Due to a favorable contribution from hydrophobic effects, stacking assumes an even greater importance in aqueous solution [50]. In addition, when the size of the indole or aromatic ring of the amino acid is similar to that of purine base, for example tryptophan, stacking may involve only one strand of the double helix RNA or DNA [35].

The types of specific interactions between aromatic amino acids and RNA or DNA are also varied. For example, as a result of the separation of the π-electrons from the nuclear charges in aromatic molecules, even if the electronic distribution is symmetric, a quadrapole electric moment arises. This allows the aromatic rings to interact as polar elements and form bonds such as π-cation. This distribution of charge in the aromatic ring confers high specificity to the bond produced between the aromatic amino acids and their cognate codons [45,47].

Possible recognition schemes for amino acids with charged residues are limited [50,51]. However, these amino acids can be bound to RNA or DNA because the α-amino and α-carboxyl groups provide good complements for hydrogen bond receptors and donors [45,50,51]. Amino acids, which can also form a pair of hydrogen bonds with particular bases of nucleic acids are Asp, Glu, Asn and Gln [51].

An interesting example of specific stereochemical affinity between nucleotides and amino acids is related to their structural similarity. In fact, a nitrogenous base of DNA or RNA can be replaced by an amino acid of similar size while maintaining the structural integrity of the nucleic acid. The nucleobase which can be replaced correlates strongly with the second base of the cognate codon [40].

A hydrophobic relation has been established between all 20 common amino acids used by life and their codons or anti-codons. Codons having U as the second base have been associated with the most hydrophobic amino acids, and those having A as the second base are associated with the most hydrophylic amino acids [52]. Some of the found associations are highly specific and may have given rise to assignations for the homocodonic amino acids (Phe, Pro, Gly, Lys) [38,51,53].

An example of how photochemical reactions could also have played an important role in the later synthesis of amino acids can be found in the proposal that two bonded nitrogenous bases could have acted as catalysts for the synthesis of simple amino acid synthesis from α-keto acids [52]. This could satisfactorily explain the hydrophobic properties of the simple amino acids and, at the same time, their affinity to their cognate codons (or at least the first two bases of the codon, a “di-codon”). The hydrophobic property would have allowed the complex to remain at the ocean surface where the UV flux was high and this hydrophobicity is inherited to the synthesized amino acid. Dinucleotides might thereby catalyze reactions required for the synthesis of amino acids by; providing free energy available through photon capture within the nucleotides [9], orientation and polarization of reactants through hydrogen bonding interactions, use of functional groups as nucleophiles or general bases, attachment of cofactors such as NADH or prebiotic equivalents, use of phosphate groups as an acid or a base catalyst, use of Mg${}^{+2}$ ions coordinated to phosphate groups as Lewis acid catalysts, and use of Mg${}^{+2}$-coordinated hydroxide ions as nucleophiles or general base catalysts [54,55,56]. UVC light interacting with nucleotides, thus probably played a fundamental role in the early synthesis of these amino acids at some point during the Archean.

In summary, there exist a strong and specific association between a number of the amino acids and their cognate codons or anticodons. Except for the proposal of Copley et al. [52] for the synthesis of the amino acids employing catalytic dinucleotides, and the direct RNA template (DRT) theory for protein synthesis proposed by Yarus [2,45], the authors are not aware of any other theory explaining the specificity of association. The rest of this article describes our proposal for the chemical–physical basis of this specificity which we relate to UVC photon dissipation efficacy and suggest a relevance to the very beginnings of the origin of life, before biosynthesis of amino acids and peptides was acquired.

## 4. Ultraviolet and Temperature Assisted Replication (Uvtar)

This section describes the proposed mechanism by which enzymeless replication of RNA/DNA-amino acid complexes could have been associated with photon dissipation. First, however, we list some of the relevant ambient conditions of the Archean Earth surface.

The temperature of approximately 80–85 ${}^{{}^{\circ}}$C of the Earth surface during the late Hadean and early Archean [57], when it is generally assumed that life arose, is strikingly similar to the short strand DNA melting temperatures, the temperature at which 50% of double strand DNA is denatured into single strands, in water at neutral pH and present ocean salinity. This is probably an important clue as to the nature of enzyme-less reproduction at the origin of life.

The temperature at Earth’s poles would have been colder, perhaps closer to the denaturing temperature of RNA under similar salt and pH ocean conditions (40 to 50 ${}^{{}^{\circ}}$C), while the temperature at the equator would have been warmer. There would also have been a temperature profile with depth and a diurnal variation of temperature in the ocean or lake microlayer similar to that of today, but probably of higher absolute temperature [21].

The Earth was cooling gradually as the greenhouse gas CO${}_{2}$ was being consumed in silicate carbonates formed through erosion of the newly forming continents [57], and once the local surface temperature fell below their respective denaturing temperatures, DNA and RNA single strand oligos (possibly formed as dissipative structures through photochemical auto-catalytic and polymerization routes [9,10,21]) would eventually find and hydrogen bond with complementary oligos and normally be unable to separate again, thus preventing subsequent template reproduction. However, through the absorption of UVC light during the day and electronic dexcitation mechanisms such as proton transfer and the dissipation of the energy of the photons into local heat, plus the absorption of solar infrared light on the ocean surface, “photon-induced denaturing” may have occurred in the late afternoon. We have quantitatively measured this UVC-induced denaturing [58,59] and it appears to be important for small oligos and to be reversible, i.e., most damage inflicted on the molecule by the UVC light is reversible and it can renature again without difficulty (see Figure 2). Such enzymeless denaturing through photon dissipation could have been helped through the association of, for example, UVC antenna amino acids, or amphipathic amino acids which kept RNA and DNA at the surface. We believe that this ultraviolet and temperature-assisted reproduction (UVTAR) mechanism is a potentially important contribution to origin of life research since enzymeless denaturing is considered to be one of the most difficult unresolved problems [60].

The characteristic steepness of the temperature denaturing curve for DNA, particularly for larger oligos and for acidic pH values ∼6.5 [61] would have facilitated photon-induced denaturing since, once the ambient temperature fell slightly below the melting temperature, the small amount of energy available in a single photon in the long wavelength UVC region (∼4.8 eV) would have been sufficient to rupture many of the hydrogen bonds between the two strands and thus separate the strands completely. Double strand RNA has a lower melting temperature and a less steep denaturing curve than DNA. During overnight periods of approximately 7 h (the Earth rotated more rapidly at the origin of life), the sharp denaturing curve of DNA would mean that the small decrease in water surface temperature would have been sufficient to allow for Mg${}^{2+}$ mediated extension of single strands [60], thereby completing the reproduction cycle (Figure 3).

Single-strand RNA and DNA is 20–40% (depending on length and sequence) more efficient at absorbing and dissipating UVC photons than double strand RNA, DNA, or the hybrid RNA+DNA duplex, an effect known as hyperchromism. It is the result of the shadowing or screening of the bases when they are tightly stacked one above the other in a native double strand arrangement [62]. In single strand RNA and DNA, the bases are free to take on arbitrary orientation with respect to the long central axis of the RNA or DNA molecule and so there is less shadowing and therefore greater photon absorption. Denaturing by UVC light would thus increase the rate of dissipation of the solar photon potential in the UVC by about 20–40% compared to native double strand. Particular RNA and DNA that could have remained in the denatured state during the day through this mechanism of UVC absorption and dissipation would have been available as single strand for template extension during the night, and thus, in this manner, be “thermodynamically” selected. If indeed this UVTAR mechanism were operative in the Archean, it would have endowed incipient life with a dissipation-replication relation, which, in fact, remains to this day in living systems, albeit now in the visible. Such thermodynamic selection would be useful for placing evolutionary theory on a non-tautological physical–chemical foundation [21].

Enzymeless photon-induced denaturing is only one half of the complete enzymeless replication problem, the other half being extension (the formation of a new strand using the single strand as a template). However, there exists experimental evidence that extension of a complimentary strand can occur at high temperatures (around 80 ${}^{{}^{\circ}}$C, similar to the surface temperature of the early Archean) using chemically activated nucleotides in an aqueous solution containing Mg${}^{2+}$ or Zn${}^{2+}$ ions [60]. There also exists experimental data indicating that enzymeless DNA extension can be sped up orders of magnitude by including the presence of planer intercalating molecules, such as the amino acid tryptophan, which prevent bending of the nucleic acid which hinders extension [63]. In our proposed UVTAR scenario, denaturing and nucleotide activation would have been induced by the UVC light during daylight hours and extension would occur overnight when the surface water had cooled by about 3–5 ${}^{{}^{\circ}}$C and there would be no UVC light inhibiting the extension (Figure 3). The ocean surface microlayer would have been rich in Mg${}^{2+}$ and Zn${}^{2+}$ ions and hydrophobic planer intercalating molecules [21].

Indications from our light-induced denaturing data [58,59], and previously published data of others concerning enzymeless extension [60], are that a rudimentary enzymeless RNA or DNA replication driven by the thermodynamic imperative of photon dissipation would have been possible utilizing day-night UVC light cycling, under the physical conditions of Earth’s ocean or lake surfaces during the Archean (high UVC flux and high temperatures with 3–5 ${}^{{}^{\circ}}$C diurnal cycling, and pH values of around 6–7).

## 5. Accumulation of Information through the Dissipation-Replication Mechanism

Being an irreversible thermodynamic process, life, since its origin, must have been coupled to the dissipation of a generalized chemical potential. We have explicitly identified this chemical potential at the origin of life in the Archean as the surface UVC photon potential (Figure 1). Furthermore, we have proposed an enzymeless ultraviolet and temperature assisted replication mechanism (UVTAR) powered by the dissipation of this light and leading to a dissipation-replication relation [7,8,9,21] (Figure 3).

In the previous sections we have emphasized how the fundamental molecules of life are pigments which absorbed strongly the UVC light available at Earth’s surface at the origin of life and throughout the Archean. We have also emphasized that the amino acids have affinity to RNA and DNA, and, in many cases, particularly to their specific codons or anticodons, which is evidence for a stereochemical era.

In the following subsections we discuss particular characteristics conferred to complexes of nucleic acids with amino acids which would promote the dissipation–replication relation and lead to greater photon dissipation. It is then shown that it is precisely these photon dissipation fomenting amino acids which have the strongest chemical affinity and specificity to their codons. Thermodynamic selection through the UVTAR mechanism (Figure 3) would then lead to the accumulation of information (codon coding) within the nucleic acids relevant to each specific characteristic promoting photon dissipation of the complex.

### 5.1. Surface Entrapment: Amphipathic Molecules

Since water absorbs and reflects UVC light (1/e extinction depth of ∼1 m) it would be crucial for optimizing dissipation and coupling to the water cycle that photochemical processes be carried out on the surface of Archean oceans and lakes where the organic molecules would have been exposed to the greatest UVC photon flux. Under normal circumstances, RNA and DNA would sediment to the ocean floor [58,59], rendering them useless for dissipation.

Eickbush and Moudrianakis [64] found that molecules intercalating RNA and DNA with an alkyl side arm which can fit into the major or minor groove could keep the nucleic acids at the surface. Even molecules like the polyamine spermidine, which are not intercalating but which can bind to the major or minor groove of DNA or RNA, thereby displacing water and making the molecule more hydrophobic, gives rise to entrapment at the surface. In fact, even the presence of simple mono- or divalent- cations saturating the primary or secondary ionic binding sites of RNA and DNA led to a low level of surface entrapment. Divalent Ca${}^{2+}$ and NH${}_{4}^{+}$, Ba${}^{2+}$ and Na${}^{2+}$ ions were found to induce entrapment at minimum concentrations about 30 times lower than that for monovalent ions. As with spermidine, the minimum cation concentration able to support entrapment corresponds to the concentration required to shield the negative phosphate charges [64].

Corroborating the suggestion that keeping RNA or DNA at the surface was fundamental to its thermodynamic function of photon dissipation, is the evidence that in the evolutionary sequence of amino acid coding, the first to be codified were most probably the hydrophobic amino acids [38,52]. The hydrophobic or non-polar amino acids are alanine (Ala), glycine (Gly), valine (Val), leucine (Leu), isoleucine (Ile), proline (Pro), methionine (Met), phenylalanine (Phe), and tryptophan (Trp). However, according to Trifonov [65], the early codon table involved only a few aminoacids; Ala, Asp, Gly, Pro, Ser, Thr and Val since these seven amino acids are encoded today by single point mutation derivatives of the presumed earliest parental codon GCU for alanine (Ala), which is a hydrophobic amino acid, as are Gly, Pro and Val [65] (Table 1). There is also a long-recognized relationship between the hydrophobicity of the amino acid and the second base of its codon. Copley et al. [52] suggest that this relationship can be explained if, before the emergence of complex biosynthetic pathways, simple amino acids were synthesized from α−keto acid precursors covalently attached to dinucleotides that catalyze the reactions required to synthesize the specific amino acid.

We suggest here that at the very beginnings of life, amphiphatic molecules (having both hydrophylic and hydrophobic parts) that can adsorb to DNA or RNA and thereby keep them afloat at the surface would have been essential to photon dissipation. The amphiphatic amino acids are; lysine, methionine, tryptophan, tyrosine (Table 1).

### 5.2. Charge Neutralization

It has been suggested that salt concentrations in the Archean ocean would have been 1.5 to 2 times greater than present day ocean concentrations due to the likelihood that salt entrapment basins at sea shores would have been limited due to late formation of continents [67] and the fact that bacterial mats preventing salt erosion back into the ocean at these sites [68] would not have existed. At the high ocean surface temperatures of the Archean, particularly near to the equator, neutralization of the negative charges on the RNA or DNA backbone by either salt ions or positively charged amino acids would have allowed overnight extension to occur at higher surface temperatures than otherwise. The positively charged amino acids would have permitted a UVTAR mechanism to be viable in environments of lower salt concentration. The amino acids histidine, lysine and arginine have net positive charge and may thus have been important to replication at higher temperatures in these low salt environments such as estuaries.

Charge neutralization of the negatively charged phosphates on RNA or DNA could be achieved by molecules which bind ionically and fit within the major or minor grooves, such as spermadine mentioned in the previous subsection or the positively charged amino acids with hydrophilic side chains. However, over saturation of the sites can lead to precipitation of RNA or DNA so there exists a narrow concentration range of such molecules supporting surface entrapment [64]. Molecules which were both charge neutralizing and hydrophobic would have performed best at keeping RNA or DNA at the surface and able to template replicate at the high ocean surface temperatures of the Archean. Of the amphiphatic amino acids (see previous subsection), only lysine is positively charged. In this regard, it is interesting the lysine is highly specifically bonded to its anticodon and is a homocodonic amino acid [38,51,53].

### 5.3. Antenna Molecules

The nucleobases of RNA and DNA have large molar extinction coefficients for light in the UVC wavelength region and are extremely rapid in the dissipation of their electronically excited singlet states to the ground state, which happens on sub-picosecond time scales through a conical intersection [10]. Also very rapid (∼2 ps) is the vibrational cooling of the hot molecule to the temperature of its water solvent environment [30].

The aromatic amino acids also have strong absorbance in the UVC region, however, their excited state lifetimes are on the order of nanoseconds, or about three orders of magnitude longer than the excited state lifetimes of the nucleic acids and they have a significant quantum efficiency for radiative decay through flourescence [69,70].

The natural stacking affinity existing between aromatic amino acids and nucleic acids keeps them close enough to permit excitation energy transfer (EET), allowing the energy of electronic excitation of the amino acid to be dissipated non-radiatively through the conical intersections of the nucleobases of RNA or DNA. Evidence of energy transfer between tryptophan and DNA exists in the observation that the fluorescence of tryptophan is completely quenched when nucleobases or nucleosides are included in solution [71,72,73,74]. The aromatic amino acid acts as a donor antenna molecule to the acceptor quencher RNA or DNA molecule, increasing the photon dissipation efficacy of the complex over what the components acting separately could achieve.

The aromatic amino acids are histidine, phenylalanine, tryptophan, and tyrosine (His, Phe, Trp, Tyr). Based on the strength of binding to their codons/anticodons, Yarus et al. [45] identify these amino acids as among the best candidates for participation in a stereochemical era based on amino acid binding at cognate codon or anticodon sites (Table 1). Contrary to the notion that these aromatic amino acids must have appeared much later in the history of life because of the complexity of their present biosynthetic pathways, UVC light offers a unique energy source for producing aromatic carbon based structures and we present evidence for this in the Discussion section.

The binding of aromatic amino acids as antenna molecules to oligonucleotides would foment the dissipation-replication relation by increasing UVC-induced denaturing. Oligos with affinity to aromatic amino acids would absorb and dissipate more photons and thus have a higher probability of UVTAR replication than random nucleotide sequences with no such affinity. In this manner the affinity between codon/anticodon and amino acid is preserved in the sequence of the codon/anticodon by what we have called “thermodynamic selection” which is acting on dissipation.

### 5.4. Intercalation

Intercalating molecules have been shown to increase the rigidity of RNA and DNA strands, thereby inhibiting cyclization which hinders enzymeless extension [75]. In fact, the efficiency for enzymeless extension in the presence of intercalating molecules can be improved by orders of magnitude [63]. To fit between the bases, these intercalating molecules must be planer aromatic and have important overlap with the bases for strongest non-covalent binding.

Tryptophan has a large UVC absorption cross section but a slow decay time of nanoseconds and high yield for fluorescence [76]. The size of its ring structure is similar to that of the bases so it intercalates strongly and can remain attached even to single strand nucleic acid [35]. Therefore, during the Archean daytime it could have acted as an antenna molecule, passing its photon-induced excited state energy to RNA and DNA through resonant energy transfer, inhibiting fluorescence and fomenting denaturing, while at night, tryptophan would have acted as an extension enhancer.

Peptides containing a planer moiety are also candidates for intercalation, for example the tripeptide LysTrpLys which intercalates through a two step mechanism; first electrostatic binding between the positively charged Lys residues and the negatively charged phosphate group of the backbone and then stacking of the indole moeity of the Trp residue [74]. Such a two step binding process has also been observed for other combinations of charged and aromatic amino acids such as LysPheArg [77]. Both of these tripeptides, with confirmed affinity to their string of cognate anticodons, retain physiological function in contemporary enzymes.

### 5.5. Catalysis

Another amino acid with a strong affinity to its anticodon is histidine [45]. This is a five-member aromatic heterocycle containing two nitrogen atoms and three carbon atoms known as an imidazole. Having two conjugate bonds, histidine absorbs strongly in the UVC (212 nm, ϵ=5700) [78] which is within the Archean UVC spectrum (Figure 1), however, it has a smaller absorption peak at 280 nm which has been attributed to photon-induced charge transfer (CT) transitions [66]. This could be important to the origin of life because imidazoles are known as amphoteric molecules, serving as both an acidic and alkaline (pKa = 14.5) catalyst for a very large number of important reactions in contemporary life, for example as a condensing agents for the phosphorylation of the nucleobases and the lipids.

The stacking of the aromatic ring of histidine with, for example, the imidazole rings of the purine bases, could lead to yet another type of catalysis by, for example, the heat generated from the photon dissipation in histidine catalyzing the subsequent thermal chemical reactions required in the UVC microscopic dissipative structuring of adenine from HCN [9,10].

Other non-aromatic amino acids that have strong absorption in the UVC Archean atmospheric window region (∼280 nm) attributed to charge transfer transitions, are the charged amino acids Lys, Glu monosodium salt (Glu·Na), Arg, and Asp potassium salt (Asp·K) [66]. According to Yarus et al. [45], the positively charged Lys and Arg, along with histidine, have strong association with either their cognate codons or anticodons (Table 1). These, therefore, by dissipating the photon energy through charge transfer, could have acted as catalysts for early biotic photochemistry such as phosphorylation of the nucleotides required for extension.

## 6. Amino Acids Which Promote the Dissipation-Replication Relation Have Greatest Affinity to Their Codons/Anticodons

In Table 1 we compare the photon dissipation characteristics (useful in the dissipation-replication relation operating through the UVTAR mechanism) of the 20 common amino acids with their specificity for their cognate condons or anticodons as determined by Yarus and coworkers [45], Johnson and Wang [46], Hendry et al. [51], and Fox et al. [53].

Yarus et al. selected eight amino acids which could have important specificity based on a review of crystallographic and NMR data for RNA-bound amino acids within riboswitches, aptamers, and ribonucleoprotein particles (RNPs). Using sequences for 337 independent binding sites directed to these 8 amino acids, Yarus et al. unequivocally demonstrate that there exists important chemical specificity for at least a subset of 6 amino acids (Ile, Phe, Tyr, Trp, His, Arg) out of the 20 common ones, to sites containing their cognate codons or anticodons. Two of the original 8, Gln and Leu, showed very weak specificity. The amino acid Lys had previously also been demonstrated to have important affinity to its anticodon [45,51,53].

Concerning the other 11 common amino acids, by determining codon or anticodon enrichment near the particular amino acid in ribosome structures of different organisms, Johnson and Wang [46] find specificity for the six amino acids identified by Yarus et al. (as well as Lys) plus seven additional amino acids, not included in the original set of Yarus et al. These seven are Met, Asp, Glu, Gln, Leu, Thr and Gly. The first three of these, Met, Asp, and Glu, have a photon disspative function in being antenna and amphiphatic (Met) or antenna and catalytic (Asp and Glu) (Table 1). Copley et al. [52] also suggest specificity of the negatively charged Asp and Glu to their anticodons of their dinucleotides, i.e., the first two nucleotides of their codon. The amino acids, Gln and Leu were assigned respectively only an extremely weak and very weak specificity to their codons and anticodons by Yarus et al. (see their Table 1), and the last two Thr and Gly were not considered for analysis, ostensibly because no localization was found for these within the RNA binding sites of the riboswitches, aptamers, or ribonucleoprotein particles studied.

We have not found any evidence in the literature for specificity to condons/anticodons of the remaining six common amino acids (Ala, Pro, Val, Ser, Cys, Asn) and this is in accordance with the lack of assignment of a dissipative function to these (see our Table 1). We can thus speculate that they, and very likely also Thr and Gly, were most probably added later to the code when more complex replication pathways were needed since the cooling of the ocean would make photon-induced enzymeless replication of RNA and DNA increasingly difficult.

It is striking that of the 10 amino acids, seven found by both Yarus et al. and Johnson and Wang to have high codon/anticodon binding specificity (Ile, Phe, Tyr, Trp, Lys, His, Arg), and three others (Met, Asp and Glu) identified by Johnson and Wang [46] and Copley et al. [52], are precisely those with the characteristics that could most enhance the dissipation–replication relation through one or more of the photon dissipation characteristics mentioned in Section 5. Furthermore, those amino acids with particularly strong UVC absorbing character were also found to have the strongest specificity to their codons/anticodons.

We suggest, therefore, that these photon dissipation characteristics are the basis of the specificity of the interaction between nucleic acids and amino acids. Optimizing the dissipation of the long wavelength UVC region of the Archean solar spectrum could thus have been the origin of the information encoding in RNA and DNA. An analogous situation is still relevant for the genomes of today’s organisms which are essentially blueprints for the construction of complex biosynthetic pathways to dissipate the external generalized thermodynamic potentials existent (or once existent) in the organisms environment.

Note that an extremely weak or very weak specificity to codons/anticodons found for the other amino acids by Johnson and Wang [46], which have not been associated with photon dissipation (Gln, Leu, Thr, Gly), would not necessarily invalidate our hypothesis of a photon dissipative basis for the specificity of the strongly stereochemical amino acids (those found by both Yarus et al. and Johnson and Wang), but may, instead, indicate that there existed some other, as yet unrecognized, and therefore probably lesser important, characteristic for fomenting enzymeless replication through photon dissipation near the origin of life which depended somewhat on these. Or, it may simply indicate that the method chosen for specificity assignment by Johnson and Wang using ribosomal proteins may not be optimal and some evidence for this is the fact that Yarus et al. in their search of riboswitches, aptamers, and ribonucleoprotein particles, did not find any specificity for Thr and Gly, and only very weak specificity for Gln and Leu.

## 7. Discussion and Conclusions

Properties of amino acids which would have been important to enhancing photon dissipation and thus proliferation of the amino acid–nucleic acid complex through the UVTAR mechanism are their (1) antenna UVC light collection for dissipation which aids in denaturing, (2) amphiphatic character required for entrapment of RNA and DNA at the ocean surface to maximize exposure to UVC, (3) intercalating facility to enhance energy transfer for rapid dissipation and for fomenting overnight extension by adding rigidity to the oligo, as well as inhibiting the production of photo-products, (4) charge neutralizing properties to foment replication at the higher temperatures of earlier times, or lower salt conditions, and (5) catalytic properties such as photon-induced charge transfer to facilitate the phosphorylation of the nucleotides required for extension, or debilitating the nucleobase stacking interaction to facilitate hydrogen bond breaking and oligo denaturing.

Yarus and coworkers [45] and Johnson and Wang [46] building on the work of others [38,51,53] have unequivocally demonstrated that there exists important specificity for 10 amino acids out of the 20 common ones to sites containing their cognate codons or anticodons, and that this is indicative of a stereochemical era near the beginning of life. Here we have shown that it is exactly these amino acids having high specificity to their codons or anticodons that have the properties necessary for fomenting an efficient photon dissipation and enzymeless replication. Furthermore, those amino acids specifically having UVC absorbing characteristics are those found to have the strongest specificity.

Since the amino acids with important photon dissipation characteristics and would have been required in enzymeless replication and since these also appear to be involved in a stereochemical era, we speculate that the first amino acids used by life were Ile, Phe, Tyr, Trp, Lys, His, Arg, and probably also Met, Asp and Glu. It is sometimes argued that the aromatic amino acids (Phe, Tyr, Trp, His) must have appeared later in the evolutionary history of life due to their contemporary complex biosynthetic pathways. However, this ignores the relatively simple UV photochemical pathways to such molecules which could have existed throughout the whole of the Archean. For example, there are relatively few steps involved in the photochemical synthesis of the even more complex nucleobase adenine starting from HCN in water [9,10]. Indeed, Cataldo [79] has suggested that the rings of the aromatic amino acids could be synthesized with UVC light acting on acetylene derived from methane and ethylene, ultimately derived from CO${}_{2}$ saturated water under UVC wavelengths [80]. Aromatic hydrocarbons have also been found in meteorites [81] and interstellar space [32,82].

We have suggested here that the first information programmed into the microscopic dissipative structures known as RNA and DNA had a direct thermodynamic utility; that of providing a physical scaffolding for the attachment of molecules which aid in the dissipation of the prevailing UVC solar photon potential. We postulate that this was achieved through a dissipation-replication relation for RNA and DNA employing an ultraviolet and temperature assisted mechanism for enzymeless replication (UVTAR) as presented in Section 4. Since this mechanism associates replication with UVC photon dissipation, it provides a non-equilibrium thermodynamic, and physical–chemical, basis for the accumulation of information relevant to the origin and evolution of life as a dissipative process.

The other amino acids without properties apparently directly relevant to the UVTAR process must have been relevant to reproduction at some point in the history of life and were probably assigned codons once metabolic pathways had been established in a more advanced coevolutionary process [6,34]. During coevolution, not only could amino acids have been added to the code, but codonic assignments may have augmented and even changed. Despite this, it is important to emphasize that there are sequences of amino acids and their codonic assignments that have remained immutable since ancestral times [83,84].

Finally, we end with a quote by Carl Woese (1967) reproduced in the seminal paper of Yarus et al. [45] referenced frequently above;

“I am particularly struck by the difficulty of getting [the genetic code] started unless there is some basis in the specificity of interaction between nucleic acids and amino acids or polypeptides to build upon.”

The work of Yarus and coworkers and of Johnson and Wang and others has demonstrated the specificity in the interaction between nucleic acids and amino acids beyond any reasonable doubt. Our work presented here is a proposal for the *basis of that specificity* which we have identified as being thermodynamic, in particular, increasing the photon dissipation efficacy (entropy production) of the RNA- and DNA-amino acid complexes which leads to higher differential replication through a UVTAR mechanism operating under non-equilibrium thermodynamic imperatives, a process we have labeled as *thermodynamic selection* [7,8,10].

## Figures and Tables

**Figure 1 entropy-22-00940-f001:**
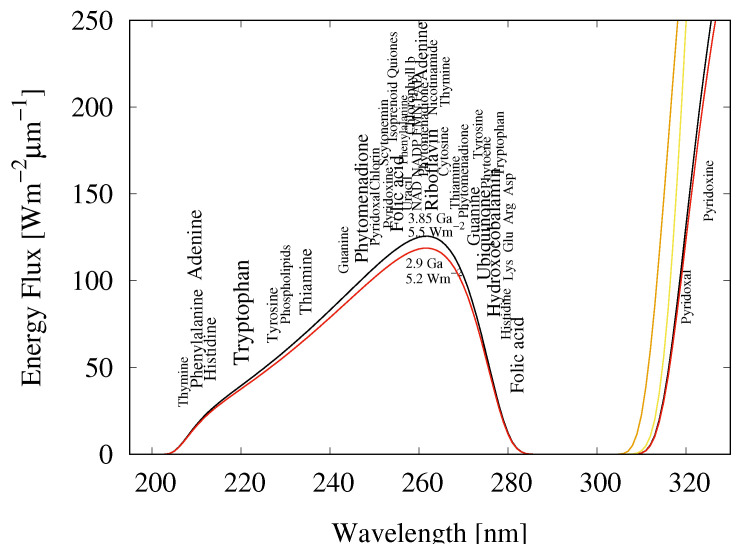
The wavelengths of maximum absorption of many of the fundamental molecules of life (common to all three domains), including the the nucleobases, aromatic and charged amino acids, cofactors, vitamins, and lipids coincide with the predicted solar spectrum at Earth’s surface in the UVC [24] at the time of the origin of life around 3.85 Ga (black line) and until at least 2.9 Ga (red line). The solar spectrum at Earth’s surface today (green line) only has wavelengths greater than about 305 nm due to absorption by ozone in the upper atmosphere. The font size of the pigment name roughly indicates the relative size of its molar extinction coefficient. Adapted from [26].

**Figure 2 entropy-22-00940-f002:**
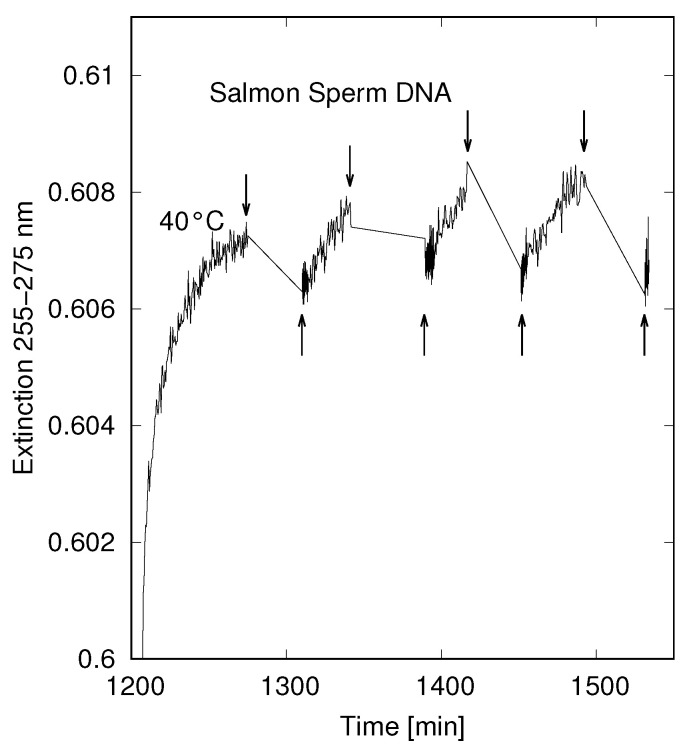
Demonstration of UVC light-induced denaturing of salmon sperm DNA of average length 100 Kbp in pure water (without salt). The temperature of the bath was raised over a time period of 20 min to 40 ${}^{{}^{\circ}}$C and later maintained at this value (±0.01 ${}^{{}^{\circ}}$C) for the duration of the experiment. The graph plots the extinction of the UVC light (due mainly to absorption with a small amount of scattering) by DNA in the wavelength range 255 to 265 nm against time as the UVC light was cycled on and off. The arrows pointing downward mark the time at which the UVC light was blocked from reaching the sample by a shutter and the arrows pointing upwards mark the times at which the light was allowed on sample by removing the shutter. It can be seen that while UVC light is on sample, the extinction increases gradually (after 1/2 h to about 0.3% of the differential absorption - between completely denatured and completely natured) due to the hyperchromicity of sections of single strand DNA arising from UVC light-induced denaturing. While the light was blocked from the sample, the segments renatured, lowering the extinction. The amount of denaturing depends on the intensity of the UVC light and the temperature of the bath. Adapted with permission from Michaelian and Santillán [59].

**Figure 3 entropy-22-00940-f003:**
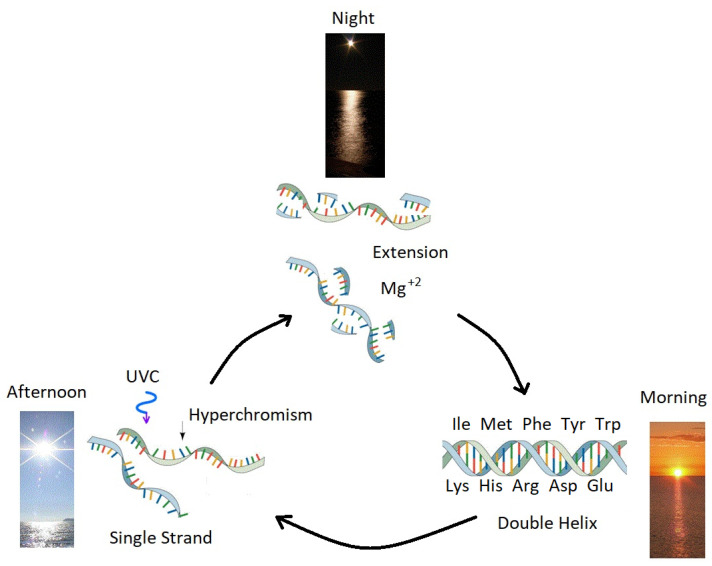
Ultraviolet and Temperature Assisted Reproduction (UVTAR) of RNA and DNA. A mechanism proposed for the enzyme-less reproduction of RNA and DNA assisted by the absorption and dissipation of the prevailing UVC light flux and the high temperatures of the ocean surface during the late Hadean or early Archean, including a day/night diurnal warming and cooling cycle of the water surface due to the absorption of solar infrared light. Most denaturing would occur in the afternoon when ocean surface temperatures were highest. Extension occurs overnight with the aid of Mg${}^{+2}$ ions, UVC activated nucleotides, and colder surface temperatures. “Hyperchromism” refers to an increase (∼35%) in the absorption of photons at UVC wavelengths (∼260 nm) once RNA or DNA are denatured into single strands. Oligos which had chemical affinity to the 10 amino acids listed in the figure (all of which have photon absorption and dissipation fomenting characteristics, see Section 5), would have had a greater chance of denaturing during daylight hours as the surface water cooled, and could therefore be replicated overnight. This selection based on greater photon dissipation we have termed “thermodynamic selection” [7,8,10,21]. The important aspect of this auto-catalytic mechanism is that replication is tied to photon dissipation, providing a thermodynamic imperative for proliferation.

**Table 1 entropy-22-00940-t001:** Comparison of the photon dissipation attributes of the common 20 amino acids, classified by hydrophobicity and by electrostatic charge, with their specificity to their codons/anticodons. Those amino acids found by Yarus et al. [45] have a strong (s; probability for non-specificity, Corr $P<10^{-4}$), moderate (m; Corr $P<10^{-3}$ ), weak (w; Corr $P<10^{-2}$), or very weak (vw; Corr $P<10^{-1}$) specificity of binding with their cognate codons/anticodons (Corr P=1 implies no specificity; see Table 1 of [45]) are identified by s,m,w, and vw respectively. Amino acids found by Johnson and Wang [46] to have specificity to their codons or anti-codons are marked with a “+” sign for association with codon/anticodon (in this case, no data is available concerning which of the homologous codons had greatest specificity to the corresponding amino acid). Lysine has also been identified as having high specificity to its anticodon [51,53]. It is striking that those amino acids with specificity also have characteristics relevant to photon dissipation, and, as a consequence, to reproduction through the dissipation-replication mechanism UVTAR. No specificity for codons/anticodons has yet been observed for the other unmarked amino acids as would be expected from their lack of photon dissipation characteristics. The “*” indicates amino acids that absorb in the 260 nm window through charge transfer transitions [66].

Amino Acid	Abbreviation	Codon	Codon/AnticodonSpecificity	Amphipathic	Antenna260 nm	Intercalating	Catalysis	ChargeNeutralizing
Aliphatic non-polar R group (Hydrophobic)
Glycine	Gly	GGUGGCGGAGGG	+/					
Alanine	Ala	GCUGCCGCAGGG						
Proline	Pro	CCUCCCCCAGGG						
Valine	Val	GUUGUCGUAGUG						
Leucine	Leu	UUAUUGCUUCUCCUACUG	+/+ /vw					
Isoleucine	Ile	AUU	s/					
AUC	+/+		yes			
AUA	/s					
methionine	Met	AUG	/+	yes	yes			
Aromatic R group (Slightly hydrophobic)
Phenylalanine	Phe	UUUUUC	/vw /+ /m		yes	yes		
Tyrosine	Tyr	UAUUAC	vw/s +/+ vw/w	yes	yes	yes		
Tryptophan	Trp	UGG	/s /+	yes	yes	yes		
Polar R group without charge
Serine	Ser	UCUUCCUCAUCGAGUAGC						
Threonine	Thr	ACUACCACAACG	+/					
Cysteine	Cys	UGUUGC						
Aspargine	Asn	AAUAAC						
Glutamine	Gln	CAACAG	/+					
R group positively charged
Lysine	Lys	AAAAAG	/s +/	yes	yes *		yes	yes
Histidine	His	CAUCAC	/vw /+ /s		yes *	yes	yes	yes
Arginine	Arg	CGUCGCCGACGGAGAAGG	vw/vw+/+/sw/s/		yes *		yes	yes
R group negatively charged
Aspartic acid	Asp	GAUGAC	/+		yes *		yes	
Glutamic acid	Glu	GAAGAG	+/		yes *		yes

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
