# Peer review of "Photon Dissipation as the Origin of Information Encoding in RNA and DNA"

_entropy, 2020, doi:10.3390/e22090940_

Round 1

Reviewer 1 Report

Thank you for providing me this opportunity to review the manuscript.

The manuscript is well arranged, and the hypothesis the authors propose is clear.

Their hypothesis is based mainly on two independent theories: one is their own (Ref. 9 and others), and the other has been published by Yarus et al. (2009) (Ref. 45).

The former theory is difficult to me to fully understand, but by my understanding, it postulates that UVC light in Archean promoted accumulation of UVC-absorbing pigments including nucleic acid bases, and that interactions between nucleobases and some amino acids could have promoted entropy production accompanying formation of dissipative structures that could support replication of the molecules involved in the structures. As the authors have already published several papers relating to the hypothesis, I do not like to point out much about its relevance. However, I could not help being suspicious if they really address physical-chemical processes during the Archean eon, the beginning of which is marked by the origin of life. I think that they mean the end of Hadean. I also wonder why they assume that the dissipative structures appeared in the ocean, not in ponds or lakes that might have been sunlit much better.

The latter theory is based on experiments. Yarus et al. have found that 8 amino acids bind well to RNA molecules that contain their cognate codon/anticodon triplet sequences.

In the present paper, the authors point out, as I understand, that the amino acids that are most likely to promote formation of dissipative structures when bound to nucleic acids are almost the same as those found to bind specifically to their cognate codon/anticodon as shown by Yarus et al. (1998). Therefore, the authors propose that the UVC photon-driven formation of dissipative structures also had some contribution to the establishment of the primitive genetic code.

I think that the hypothesis is interesting and should be published in some form or another, irrespective of its flaws, if the authors can answer properly to the major question as follows.

The major problem of the hypothesis, I think, resides in that Yarus et al. (2009) do not state that the eight amino acids are all that bind to their cognate codon/anticodon. According to my understanding, they just selected nine amino acids that had been a target of an experiment for selection of RNA molecules that have a high affinity for the amino acid. Eight out of the nine binds specifically to RNA molecules that contains the sequence of their cognate codon or anticodon. Thus, the other 11 amino acids may also bind to the cognate codon/anticodon trinucleotide.

On the other hand, the major ground of the present theory is that the amino acids that are most likely to promote entropy production coincides with the amino acids that have been shown to be strongly connected to the codon/anticodon triplet. If one or more of the 11 amino acids also binds specifically to either of the codon and anticodon triplets, then the essential ground of the theory is lost.

Therefore, I recommend that the manuscript be published only if the authors can solve this major problem.

The text is readable, though there are several misspellings.

Author Response

We thank the reviewer for their work reviewing the manuscript and for their comments and questions which have helped produce an improved manuscript.

The understanding of our article that the reviewer provides in their review summary is indeed what we had intended to portray. Specifically (quoting the reviewer);  “that the amino acids that are most likely to promote formation of dissipative structures when bound to nucleic acids are almost the same as those found to bind specifically to their cognate codon/anticodon as shown by Yarus et al. (1998)”.

In agreement with the reviewer, we believe that these molecular structuring processes could have occurred during the late Hadean as long as surface temperatures were not too high to cause molecular instability. The beginning of the Archean is considered to mark the beginning of a stable Earth surface, just after the period known as “the late lunar bombardment” which could have caused periodic sterilizations of Earth’s surface. However, as the reviewer states, there is no in principle reason why such molecular synthesis and complexation could not have occurred during the later part of the Hadean. We have therefore amended the text to reflect this.  

The reviewer is also correct in that these processes could have also occurred on the surface of lakes and ponds, besides the ocean surface. This would, in fact, allow for even more varied conditions of pH, salt, and temperature and temperature cycling. However, it is generally believed that most formation of continents occurred later in the Archean, about 3.6 Ga [Cheney, 1996; Knauth, 2005], and that before this, shallow seas covered much more surface area than the deeper oceans existing today.  If these processes occurred in lakes, then oligo denaturing temperatures would be lowered since the salt concentration of lakes would probably be lower than that of the shallow seas. This would push forward the beginning of the enzymeless oligo replication period (assuming the validity of some type of UVTAR mechanism) later into the Archean when surface temperatures became colder.  In the revised version of the manuscript, we have included lakes and ponds as potential surfaces for these processes.

Yarus et al. selected 8 amino acids which could have important specificity for their codons/anticodons based on a review of crystallographic and NMR data for RNA-bound amino acids within riboswitches, aptamers, and ribonucleoprotein particles (RNPs). Using sequences for 337 independent binding sites directed to these 8 amino acids, Yarus et al. unequivocally demonstrate that there exists important chemical specificity for at least a subset of 6 amino acids (Ile, Phe, Tyr, Trp, His, Arg) out of the 20 common ones, to sites containing their cognate codons or anticodons. Two of the original 8, Gln and Leu, showed very weak specificity. The amino acid Lys had previously also been isolated as having important affinity to its anticodon [Yarus et al., Fox et al., 1971, Hendry et al. 1981].

Concerning the other 11 common amino acids, we took up the challenge put to us by the reviewer and performed an extensive literature search, finding only two more papers with useful data on the specificity of amino acids to codosn/anti codons. One by Copley et al. (2005) which showed some evidence for specificity of negatively charged Glu and Asp to their anticodons of their dinucleotides (the first two nucleotides of their codon), and a paper by Johnson and Wang (2010) in which they similarly find the 6 amino acids identified by Yarus et al. and Lys, plus 7 additional amino acids, not included in the original set of Yarus et al., which apparently show some specificity to their codons or anti-codons as determined by codon or anticodon enrichment near their respective amino acids in ribosome structures of different organisms. These 7 amino acids are Met, Asp, Glu, Gln, Leu, Thr, Gly (see data in section D and E of Fig. 1 of Johnson and Wang, 2010).  Concerning the first 3 of these, Met, Asp, and Glu, we had already assigned a disspative antenna and amphiphatic function to Met, and an antenna and catalysis function to Asp and Glu, so this new data from these two papers provided us with further support for our thesis. The amino acids, Gln and Leu were assigned only an extremely weak and very weak, respectively, association to their codons and anticodons by Yarus et al. (see their table 1) and the last two Thr and Gly were not even considered by Yarus et al., ostensibly because there was no association found for these to the binding sites of riboswitches, aptamers, or ribonucleoprotein particles studied.

We have not found any evidence in the literature for specificity to their condons/anticodons for the remaining 6 common amino acids (Ala, Pro, Val, Ser, Cys, Asn) and this is in accordance with the lack of assignment of a dissipative function to these (see our Table 1). We can thus speculate that these, and most likely also Thr and Gly were most probably added later to the code when more complex replication pathways were needed since the cooling of the Earth’s surface would make photon-induced enzymeless replication of RNA and DNA increasingly difficult.

An extremely weak or  very weak association of Gln and Leu to their codons or anticodons does not necessarily invalidate our suggestion of a photon dissipative basis for the specificity of the strongly stereochemical amino acids (those found by both Yarus et al. and Johnson and Wang), but may, instead, indicate that there existed some other, as yet unrecognized, and therefore probably lesser important, characteristic for fomenting enzymeless replication through photon dissipation near the origin of life which depended somewhat on Gln and Leu. Or, it may simply mean that the method chosen for specificity assignment by Johnson and Wang et al. using ribosomal proteins may not be optimal, and some evidence for this may be that Yarus et al. in their search of riboswitches, aptamers, and ribonucleoprotein particles did not find any specificity for Thr and Gly and only very weak specificity for Gln and Leu.

One final point in support of our proposal, regarding the specificity of amino acids to their codons/anti codons and the photon dissipative functions assigned to them; those amino acids with particularly strong UVC absorbing character are also those found to have the strongest specificity to their codons/anticodons.

We have included the two new references with the new evidence of specificity in the revised version as well as three new paragraphs in section 6 in response to the reviewer’s challenge. We have included a new category of “very weak” (vw; Corr P<10-1) association of binding sites with their cognate codons/anticodons in table 1 of the revised manuscript and we have updated the table to include the new information from the Johnson and Wang reference. We have also combined the Discussion section with the Conclusions section.

We thank the reviewer, particularly for presenting us with this challenge concerning specificity of the other amino acids not identified by Yarus et al., since it led to our discovery of the Copley et al. and Johnson and Wang papers and the realization of an assignment of specificity for the UVC absorbing amino acids Met, Glu and Asp, which were previously (in the prior version of the manuscript) without an assignment of specificity.  

We have corrected a number of misspelled words, made numerous small changes to improve the redaction of the text, updated Table 1 in accordance with the above, and include an improved version of Fig. 3.

References;

Knauth, L.P. Temperature and salinity history of the Precambrian ocean: implications for the course of microbial evolution. Paleogeography, Paleoclimatology, Paleoecology 2005, 219, 53–69.

Cheney E.S.  Sequence stratigraphy and plate tectonic significance of the Transvaal succession of southern Africa and its equivalent in Western Australia". Precambrian Research. 1996, 79 (1–2), 3–24.

Copley, S.D.,  Smith, E.,  Morowitz, H.J. A mechanism for the association of amino acids with their codons and the origin of the genetic code. PNAS 2005, 102, 4442–4447.

Fox, S.,  Lacey, J.J., Nakashima, T. Nucleic Acid-Protein Interactions, D.W. Ribbon and J.F.Woessner 1971, p. 113.

Hendry, L.B.; Bransome, E.D. Are there Structural Analogies Between Amino Acids and Nucleic Acids? Origins of Life 1981, 11, 203–221.

Johnson, D.B., Wang, L. Imprints of the genetic code in the ribosome. Proc Natl Acad Sci U S A. 2010, 107(18):8298-303. doi: 10.1073/pnas.1000704107.

Reviewer 2 Report

Living systems from the point of view of thermodynamics of irreversible processes are dissipative structures, that is, systems in which the production of entropy exceeds the production of entropy in the environment. Until now, it was not clear why such structures were formed, which subsequently led to the emergence of life on Earth. The authors propose and logically substantiate on the basis of modern molecular biology data that solar energy is the main factor that increases the production of entropy in molecular systems. Such an understanding of the formation of prebiological systems makes a significant contribution to the understanding of the thermodynamic foundations of the emergence of living systems. The peer-reviewed article is extremely interesting and will be useful for biologists, regardless of their specialization.

Author Response

We thank the reviewer for their kind remarks concerning our manuscript and for their recommendation to accept for publication.

Round 2

Reviewer 1 Report

I think that the manuscript is almost ready to be published. However, there are several minor issues remaining.

Lines 330-331
I think that the sentence "Divalent ... for monovalent ions." can be removed. In this context, information about the relative ability of the divalent ions as compared to that of the monovalent ions is not needed. Alternatively, it may be better to mention that Ref. 64 suggests that small molecules that partially neutralize the phosphate charge and simultaneously add some hydrophobicity to DNA molecules are likely to cause entrapment of the DNA molecules arount the air-water surface.
If the sentence is needed, authors should consider that, according to Ref. 64, divalent Ca2+ and Ba2+ ions induce the entrapment at minimum concentrations about 30 times lower than that for monovalent NH4+ and Na+ ions. ... The divalent ions are Ca2+ and Ba2+, and the monovalent ions are NH4+ and Na+. Na2+ does not exist.

The 7th line in the Figure 3 caption, Lines 214 and 215.
"Mg+2" should be "Mg2+".

Lines 350-351.
Why these 5 amino acids are amphiphatic and the others not? In particular, why Arg is not amphiphatic?

This manuscript is a resubmission of an earlier submission. The following is a list of the peer review reports and author responses from that submission.

Round 1

Reviewer 1 Report

The authors have written an excellent paper drawing on the literature to answer the question that Carl Woese posed as Origin of Life's greatest challenges. To explain how amino acids developed their specific affinity to their codons in the genome.  The authors tell us at the start of the paper that they are using Stereochemical Theory within a framework of Non-equilibrium Thermodynamics Theory. 

The basic thesis is that amino acids were favored because they efficiently dissipated the energy from UVC light circa 260nm. Figure 1 shows the grouping of these molecules around the 260nm peak. They were selected thermodynamically because they increased the rate of entropy production.

line 149  187: The authors give 8 distinct types of association between amino acids, nucleotides and their nucleic acids. They explain why four give rise to specificity and the other four do not. I found this section especially helpful.

Figure 2 was useful in showing the concept of how DNA could denature under light  and reform once the light had been blocked. Figure 3 shows the daily cycle of DNA and RNA  being broken into single strands and then overnight reform the double strand form. For me this strengthened their argument.

I think that the authors have provided insights into possible pathways during the Origin of Life that produced specificity of amino acids to their codons and anticodons, and their replication. 

Given the nature of the interactions involving replicating molecules, I was wondering what role Dynamic Kinetic Stability would play in the Archean Seas in the Origin of Life? The turnover of replicators could provide a direction of evolution for the system? Certainly the processes which replicators experience are similar to the processes that organisms (single or multicellular) experience in evolution, e.g. Natural Selection. You may reflect on that and consider it in another paper, if you feel it is worth pursuing.

Overall, I found the paper well structured, well argued and clearly presented. I am enthused by this novel idea of how the specificity of amino acids and their codons evolved.

Author Response

We thank the referee for their work in reviewing the manuscript and for their kind and encouraging comments which provide us with great motivation.

Regarding the question concerning dynamic kinetic stability, we would like to inform the referee that we are at this moment finishing a paper on this very topic. Our approach is through classical irreversible thermodynamic theory in the non-linear regime. For systems of chemical or photochemical reactions with positive feedback (catalysis) there are numerous stationary states available to the system, some are dynamic (rotations about the stationary state) in the intermediate product concentrations and some of which are spatially inhomogeneous in the concentrations. The stability of each stationary state has to be analyzed separately using the Glansdorff-Prigogine universal evolution criterion d_XP<=0. There is no total differential that can be used as a potential to describe the evolution of the system, but the Glansdorff-Prigogine criterion can be used to show that those stationary states of greater concentration of catalyst product will be those most likely to be the targets of evolution through amplification of a microscopic fluctuation which takes the system out of its region of local stability and into another. Such stationary states which are the targets of this kind of thermodynamic evolution have greater concentrations of catalyst and thus dissipate more rapidly the imposed chemical, or photochemical, potential. The referee may be interested in our response to reviewer 2 since in this response we go into more details about the classical irreversible thermodynamic theory which we base our paper on.

Replicators having characteristics endowing them with greater photon absorption and dissipation into heat, would be those more likely to denature during the day as the ocean surface temperature was cooling and these then would then be replicated overnight so this would imply that evolution would be directed towards greater dissipation.

We have included three new paragraphs in the “Foundations” section, section 2, beginning at line 63, of the revised version of our manuscript which summarizes the above.

Reviewer 2 Report

M. Mossio, Montévil, M., and Longo, G., “Theoretical principles for biology: Organization”, Progress in Biophysics and Molecular Biology, vol. 122, no. 1, pp. 24-35, 2016

The paper "Entropy Production as the origin of information encoding in RNA and DNA" aims to provide a plausible scenario for the appearance of the genetic code. I am not a chemist, therefore i cannot really asses the chemical discussion per se. 

There are some pitfalls in the use of entropy production and classical irreversible thermodynamics in the paper.  This framework requires special conditions to be met (local thermo equilibrium) and it is not a general framework for phenomena far from thermodynamic equilibrium - even among the inert. For example, the following sentence is wrong because it confuses this specific framework with being far from thermodynamic equilibrium.  

 "This organization of material is known as "dissipative structuring” in non-equilibrium thermodynamic language. Biologists are acutely aware of a sub-set of this structuring and refer to it simply as“life"." l54-55

Many people have argued that this framework is inadequat or insuufficient for biology, for example 

G. Longo, Montévil, M., Sonnenschein, C., and Soto, A. M., “In search of principles for a Theory of Organisms”, Journal of Biosciences, pp. 1-14, 2015

Kauffman, S. A. (2019). A World Beyond Physics: The Emergence and Evolution of Life (Oxford University Press)

l101 

"Since microscopic dissipative structuring can be persistent, i.e. structuring remains even after the removal of the impressed external potential, information concerning the impressed potential and the molecular structuring required for its dissipation becomes programmed into the structure of the material."

This statement lack accuracy. It is true to an extent, but only for a limited time scale. For example, DNA life time in standard condition is several centuries, therefore the persistence of dna patterns at evolutionary requires the active maintenance by biological activities, see the discussion in 

M. Mossio, Montévil, M., and Longo, G., “Theoretical principles for biology: Organization”, Progress in Biophysics and Molecular Biology, vol. 122, no. 1, pp. 24-35, 2016

 Overall, i strongly suggest to argue why, in this case, there is an optimization of entropy production -if any- and at what level. It seems that there is "competition" between different molecular variants", but does it genuinely entail the optimization of entropy production -maybe here but not in biology in general since free energy is used for structuring biological organizations and  its use can be deferred. 

Last, the text is well written but there are many typos.

Author Response

We thank the reviewer for their work in reviewing our manuscript, and for giving us the opportunity to publish in this special edition of Entropy.

We are aware of the general perception among researchers that Classical Irreversible Thermodynamic theory is not a good theory on which to base the description of biological phenomena because it is based on the predicate of “local thermodynamic equilibrium” whereas biological phenomena are considered by these to be too “far from thermodynamic equilibrium” to be treatable under this theory.  However, the existence or not of “local equilibrium” must be determined in every situation where the application of CIT theory is contemplated. Most life processes are based on chemical reactions and diffusion processes in condensed material. Both these processes involve interactions (collisions) between molecules, or between molecules and ions. For a chemical reaction to be treatable under CIT theory, it is only required that a Maxwell-Boltzmann (equilibrium) distribution of the molecular or ion velocities be maintained [1]. In condensed materials, such as biological material, this is guaranteed by the extraordinarily large numbers of interactions which occur on time and spatial scales much smaller than that of the observable macroscopic phenomena being studied.  Chemical reactions, except for very explosive reactions (which rarely occur in biological systems), can thus be safely treated within the CIT framework. In fact, the majority of examples in Prigogine’s seminal book “Introduction to the Thermodynamics of Irreversible Process” are chemical reactions.

Diffusion processes in condensed media similarly involve extremely large numbers of interactions on very short time scale such that local equilibrium is guaranteed, except where particle densities are very low (which never occur in biological systems) such as in outer space.

The assumption of local equilibrium for our particular case presented in the manuscript of molecular photochemical dissipative structuring under the UVC photochemical potential requires that the absorbed energy (of the photon) becomes distributed with Boltzmann statistics (equilibrium) over the nuclear vibrational degrees of freedom responsible for molecular transformations. Organic materials in the condensed phase are generally soft materials in the sense that their electronic degrees of freedom couple significantly to their nuclear vibrational degrees of freedom (unlike in the case of inorganic material). This nonadiabatic coupling is mediated by conical intersections which allow for ultra-fast equilibration of the photon energy over vibrational degrees of freedom of the molecule, often on femtosecond time scales, leaving the molecule with an effective vibrational temperature of 2000-4000 K after the absorption of a single UVC photon. This time for vibrational equilibration is generally less than the time required for a typical chemical transformation and therefore the irreversible process of molecular dissipative photochemical structuring can be justifiably treated under the CIT framework in the non-linear regime. A summary of this argument has been included in the revised version of the manuscript.

Regarding the referees concern about the optimization of the entropy production, CIT theory suggests that in the non-linear regime there does not exist a general potential (total differential) whose optimization describes the evolution of the system. Only in the linear regime does the entropy production become a thermodynamic potential whose minimization, in fact, describes the evolution of the system toward a unique stationary state.  In the non-linear regime, however, there does exist a non-total differential with a definite sign which can be used to determine aspects of the evolution of the system, such as its local stability in the given stationary state and its statistical probability of leaving that stationary state and evolving to another neighboring state (in the regime of non-linear relations between forces and flows more than one stationary state becomes available to the system). This is known as the “general evolution criterion” or the “Glansdorff-Prigogine” criterion.  It states that the variation of the entropy production with respect to the rearrangement of the free forces (those not fixed externally) must be negative semi-definite, i.e. dXP<=0, and this is valid in the whole domain of applicability of CIT theory (local equilibrium and constant external constraints). For chemical reactions, the free forces are the free affinities divided by the temperature. There is, however, no restriction on the evolution of the total entropy production dP=dXP+dJP since this also includes a contribution due to the rearrangement of the free flows dJP (the flows being the rates of the chemical reactions). The total entropy production during the evolution may either increase or decrease depending on the nature of the particular system.   

In chemical systems (inherently non-linear) in which there is positive feedback, for example, when one of the products of a reaction acts as a catalyst for some part of the overall reaction, then numerous stationary states may exist and the different states may be reached from a given state by the amplification of a fluctuation. In this case, each stationary state has to be individually evaluated for its local stability. In these systems, application of the Glansdorff-Prigogine criterion leads to the interesting result that evolution, through amplification of fluctuation, leads to stationary states of greater dissipation (see the example in Glansdorff and Prigogine [2]). The non-linearity of the positive feedback (catalytic activity) means that microscopic fluctuations which temporarily remove the system from the local stability surrounding a stationary state are amplified and generally lead the system to a new stationary state in which the concentration of catalyst product in the system has increased, and therefore the overall reaction rate (and thus the overall entropy production) has also increased.

There are many examples of evolutionary increases in dissipation in biology; from the increase in metabolic rates per unit biomass over evolutionary history, to the increase in photon dissipation rates of ecosystems as they go through succession to reach their climax state. In our particular case, of the complexation of DNA with particular amino acids, it is the rate of UVC photon dissipation that is optimized.

We have included three new paragraphs in the “Foundations” section, section 2, beginning at line 63, of the revised version of our manuscript which summarizes the above.

We have removed the sentence “Biologists are acutely aware of a sub-set of this structuring and refer to it simply as``life".”

At least at the initial stage of life or which our paper is concerned, it is not competition between molecular configurations for survival, (e.g selection of greater chemical stability) but rather the selection of the distribution of concentrations of the dissipatively structured molecular configurations (corresponding to a particular non-equilibrium thermodynamic  stationary state) which improves the dissipation of the incident photon flux.  These non-equilibrium distributions are dependent on the incident photon flux for their structuring, proliferation and evolution and such evolution is only “concerned” with improving the dissipation of the incident photon flux.

We believe that this is true also today and at all hierarchal levels up to the biosphere. However, it is much more difficult to see how dissipative structuring is being thermodynamically selected at these levels since this involves an enormous amount of nested, coupled, and interdependent dissipative processes (including abiotic ones such as the water cycle) and this thermodynamic selection for dissipation occurs on the entire system.  In an article previously published [3] I presented a simple model in an attempt to make the ecosystem amenable to this type of thermodynamic analysis with selection operating under thermodynamic dictates of the Glansdorff-Prigogine evolution criterion and acting at the global level (on the total entropy production of the ecosystem) by treating species as if they were made up of a population of units of entropy production and exchange and I was able to show how population dynamics similar to ones observed in nature emerge without the necessity of introducing any kind of competition between individuals or between species. The system selects those stationary states consisting of concentration profiles (sets of species populations) which best dissipate the incident free energy flux.

Some of the free energy in the photon flux does indeed go into structuring, on both the microscopic and macroscopic scales, and some of this free energy can be stored, but this is only a very small part of the total dissipation performed by the system. For example, of the free energy absorbed by plant leaves <1% is channeled into photosynthesis, the rest is simply dissipated directly into heat.

Concerning the persistence of the dissipative structuring, the particular programming of the genome will persist until it is no longer being selected for by the external photon potential (e.g. because of a change in the wavelength intensity distribution, or because of a more efficient dissipative structuring is found, or because life may become attached to a completely different generalized thermodynamic potential, e.g. a chemical potential). Certainly the persistence is not maintained over evolutionary time scales, not least because of the fact that the external photon potential that we assumed initially synthesized nucleic acids and amino acids disappeared from Earth’s surface about 1,000 million years after the origin of life. A sentence beginning at line 148 has been included in the revised version of the manuscript to address this.

We have corrected spelling mistakes and other typos and have improved the redaction in a few places.

Finally, we have changed the title somewhat to emphasize our suggestion that it was UV photon dissipation that was responsible for the dissipative structuring and the first acquisition of information.

References:

[1] I. Prigogine. Introduction to Thermodynamics Of Irreversible Processes. John Wiley & Sons, third edition, 1967.

[2] P. Glansdor_ and I. Prigogine. Thermodynamic Theory of Structure, Stability and Fluctuations. Wiley -

Interscience., 1971.

[3] Michaelian, K., Thermodynamics Stability of Ecosystems, J. Theo. Biol., 237 (2005) 323-335.

Reviewer 3 Report

The authors' point of view is quite justified and has the right to exist. Therefore, I believe that the peer-reviewed article should be published, despite the fact that I do not agree with certain provisions of the authors. In my opinion, the production of entropy is not a source of any processes, but their consequence. The appearance of dissipative structures is rather associated with conjugate processes, as a result of which energy is released. This energy is spent on maintaining the level of dissipative function in dissipative structures. An increase in the entropy of the system as a result of its production leads to the fact that the level of disequilibrium of the system as a whole decreases. Therefore, the production of entropy cannot serve as a source of the formation of dissipative structures.

Author Response

We thank the reviewer for their work in reviewing the manuscript and for their recommendation to publish our paper.

As Prigogine has shown, a system may either increase or decrease or remain the same in its entropy production on leaving one stationary state and evolving towards another. Therefore, in agreement with the reviewer, within the validity of the framework of classical irreversible thermodynamics, entropy production is not a universal potential and we cannot assume that it “drives” the evolution of any arbitrary system.  The only criterion we have that could be imagined to “drive” the system is the Glansdorff-Prigogine evolution criterion which says that the change of the entropy production with respect to the changes in the free forces is negative definite.  Now, it turns out that for purely dissipative (no convection) auto-catalytic systems (positive feedback) the Glansdorff-Prigogine criterion leads to the result that the total entropy production for these systems does increase through time under this criterion through amplification of fluctuation at bifurcations (see the example of Glansdorff and Prigogine starting on page 272 of their book [1]).  For these particular systems there is a tendency of nature to increase dissipation and it may be a matter of perspective to state that it is entropy production which drives that system, in reality, it is the microscopic fluctuations being selected through the Glansdorff-Prigogine criterion at the foundation, but both give the same result for these particular systems. We have been more careful in the revised manuscript to make this clear.

We note also that although these systems increase their internal entropy production , d_iS/dt, the entropy of the system itself does not necessarily increase since there is always a net flow of the entropy to the environment, d_eS/dt.  It is this flow of entropy that maintains the stationary state (i.e. dS/dt = d_iS/dt +d_eS/dt = 0 for the system in a stationary state).

[1] P. Glansdorff and I. Prigogine, Thermodynamic Theory of Structure, Stability and Fluctuations, Wiley-Interscience, New York, 1971.

Reviewer 4 Report

Authors of the paper: „The Origin of Information Encoding in RNA and DNA through UV Photon Dissipation“ provide a qualitative description for their hypothesis about the enzyme-less origin of RNA and DNA at the surface of Archean oceans almost four billion years ago. They assume that UVC solar photons were the driving force for the synthesis of nucleic acids and amino acids.

I must first point out that the paper is classified as an „Article“ and my review is written with that assumption in mind, namely, that the paper is submitted to Entropy journal (authors note that it is submitted to „Journal Not Specified“) as regular research paper, not as an invited review paper. However, the paper is not written as an original research paper. It does not have the Results section, nor the Materials and Methods section. It is written as a review paper or hypothesis paper. Such paper types can have some new original theoretical or experimental results too. However, there are no new theoretical or experimental results in this paper.

Plainly speaking, the common leitmotif of the paper is confusion. Confusion between research type paper and review type pape, confusion among an extraordinary large number of unsupported speculations (some even contradictory to other mentioned speculations), confusion among present-day biochemistry and imagined events at the Archean era, confusion between logical and circular reasoning on top of confusion in using English language („synthesis of synthesis“ a the line 240), and the most bothersome confusion among what is original in this paper when compared with other published papers by same or other authors.

I shall first focus on figures 1-3 and Table 1 because these are all figures and tables from the paper and a reader would expect to see some novel and original results there. 

Figure 1 about the absorption of biological chromophores in the UVC was already published by the second author in his two earlier papers (references 9 and 26). Some unspecified adaptions have been performed with regard to figure 3 of reference 26 without mentioning identical figure 1 from reference 9.

Section 4 about the UVTAR hypothesis presents in figure 3 the proposed mechanism for UV and temperature assisted reproduction of RNA and DNA with the abbreviation UVTAR as if it is a new proposal. However, the identical hypothesis and abbreviation were used by the second author in his 2011 paper (reference 8). This is obscured by the authors' idea to use the term „microscopic dissipative structures“ for RNA and DNA and with their mentioning of reference 8 only in the introductory section.

Figure 2 has been copy-pasted from the second author's 2019 paper in the Heliyon periodical (reference 57, Figure 10) without any adaptation that I can see. However, authors claimed that they performed some (unspecified) adaptation by using the grammatically incorrect sentence in the figure 2 legend: „Adapted with permission from Michaelian and an [57]“.

Table 1 is first presented as a new suggestion (result?) for binding of some amino acids to their codons or anticodons but then at rows 420 to 422 it becomes clear that these are 10 years older results from Yarus et al. paper in 2009 (reference 44) with some added amino acid features by present authors. Unfortunately, Yarus probability values and all explanations about them are omitted so that fourth column with s, m, w classifications and Corr P inequalities (introduced by authors) becomes completely obscure and I would say meaningless as well. Furthermore, the author's classifications of glycine as a hydrophobic amino acid, histidine as positively charged amino acid (at neutral pH?), threonine as a not-amphipathic amino acid, and cysteine as a not-hydrophobic amino acid is questionable without being supported by any arguments.

Expectations to see some original and well supported new results were not fulfilled. Authors do mention „we present evidence for this in the Discussion section“ (evidence how UVC light offers a unique energy source for producing aromatic carbon-based structures) at the 425 as if it is a standard scientific method to postpone any hard evidence until the Discussion section. However, like almost all other statements throughout their paper, what we can find in the Discussion section is not the evidence but are only conditional mood statements: „may have been important“, „could have existed“, „it is probable“, „could be derived“, „were probably assigned“. These conditional statements are also repeatedly used to cite older references instead to discuss pro and contra arguments about the author's original findings in this paper (if any). It all boils down to repeating earlier proposals or hypothesis how life could have originated at a warm ponds or at very thin layer of surface ocean water as if such proposals have been never criticized and as if they are not contrary to most other recent theories about life origin that are not even mentioned in the present paper.

The universal evolution criterion of Glandsdorff and Prigogine is used by authors only as a window-dressing to impress readers without any invested effort to prove its relevance for photon-dissipation driven autocatalytic reactions presumably leading to nucleic acid enzyme-less replication and extension.

To sum up this part of the review about presented figures and tables, in my opinion, none of them should have been presented. Only table 1 may indicate something interesting since it mentions some Corr P inequalities in its legend. However, few „Yes“ and many empty spaces presumably meaning „No“ or „Not“ cannot be used to make any correlation and even worse are not author's results but confused qualitative rendering of Yarus et al. results (reference 44) with two out of four interesting amino acid residues (for „dissipative-replication“) confusingly belonging to both strong (s) and weak (w) category.

The lack of explanations and critical thinking is another common leitmotif of this paper.

There is no explanation of why the question of carbohydrate and lipid synthesis is completely neglected when the origin of life is discussed. Nucleotides are mentioned many times but not their ribose or deoxyribose components. Amphipathic polar lipids are crucial to forming first bioenergetic membranes able to maintain higher inside concentrations of proto-catalytic organic molecules in topologically closed membranes but lipids are not mentioned in this context at all. There is no acknowledgement, much less any proposed explanation, that life uses only left-handed (L) amino acids and D-sugars (D-ribose). There is no mention that electric sparks or UV radiation produce only the racemic mixture of some amino acids. Chirality is mentioned only in the first paragraph from the introductory section. The paper does not describe experimentally reproduced mechanisms for the synthesis of nucleotides and aromatic amino acids by using UV day-night temperature cycling. There is no explanation of why present-day DNA (salmon sperm DNA) denaturation is considered relevant for the origin of nucleic acids. No technical details are given about these DNA samples. There is no Methods section in the paper probably because no experiments were performed specifically for this paper. There is no explanation of why DNA and RNA origin are not distinguished and much more versatile „RNA-world“ first hypothesis is not mentioned et all. The possible and likely role and identity of proto-enzymes is also not mentioned. There is no explanation why peptides are not considered important for the origin of life when it was mentioned (line 258 and 259) that peptides and biosynthesis of amino acids is probably less relevant for the very beginnings of the origin of life. There is no explanation of how trinucleotides can be synthesized without enzymes despite the focus of the paper on the binding of some amino acids to codons or anticodons. There is no discussion on why UV is regularly used today to kill microorganisms and why it is important to maintain our ozone layer in the atmosphere as the protection against DNA-damaging effects of UV radiation. This seemingly contradicts the postulated ability of UVC to create nucleic acids. There is no explanation or discussion when sedimentation can start to be the problem.

Some specific remarks are in order too:

Line 95 – Non-linearity is not due (and not only due) to the feedback effect.

Line 140- Circular reasoning: Authors assume that nucleic acids existed in the Archean era as the proposed explanation for the photon-induced synthesis of nucleic acids

Line 171-Grammatical mistakes even in titles of sections

Lines 307-318 and 434 – Confusing extension with replication

Lines 362-373 – Confusing and contradictory statements about which amino acid types were the most important for the origin of life: hydrophobic or amphipathic

Lines 426-431- Would-would sequence about „thermodynamic selection“ proposal.

In summary, instead of proposing just one novel and original hypothesis with as much theoretical and experimental evidence provided by authors as possible, we have got in the submitted paper an overwhelming number of unsupported speculations together with a lack of clear logical thinking.

Author Response

We thank the reviewer for their work in reviewing our manuscript.

The reviewer presents a number of different lines of criticism with similar points within a given line but associated points are dispersed throughout his/her report. We therefore provide a rebuttal to each of those lines of criticism as a group.

1) The reviewer, in numerous instances, suggests that we have published everything before and that there is nothing new in our paper. The reviewer states, “However, there are no new theoretical or experimental results in this paper.”

This is categorically untrue and unfair. In the introduction and conclusions of our article we emphasize that our manuscript is an attempt to answer the question posed by Carl Woese in 1967, which until now has remained unanswered, i.e. “…the difficulty of getting [the genetic code] started unless there is some basis in the specificity of interaction between nucleic acids and amino acids or polypeptide to build upon.” Our paper specifically identifies a basis for this specificity, a thermodynamic basis, that of increasing photon dissipation efficacy of the RNA- and DNA-amino acid complexes.  And this, we believe, is the origin of information encoding in DNA and RNA. This is a novel assertion and our paper presents the evidence for this based on the analysis of published data. This proposal for resolving the problem has not been published in any previous paper.

2) The reviewer suggests that we have left many explanations out of our article concerning the origin of life, for example, he/she states “There is no explanation of why the question of carbohydrate and lipid synthesis is completely neglected when the origin of life is discussed.”  and “There is no acknowledgement, much less any proposed explanation, that life uses only left-handed (L) amino acids and D-sugars (D-ribose).” He/she then goes on to mention another 4 aspects of the origin of life that, according to him/her we should have included.

It is unreasonable for the reviewer to expect us to present a complete theory on the origin of life in our paper.  Our paper is only dedicated to providing a plausible answer, along with the supporting empirical data, to the question posed by Woese of what could have been the basis of the specificity in the interaction between nucleic acids and amino acids at the origin of life. We provide a non-equilibrium thermodynamic explanation based on optimizing photon dissipation, which is at the foundation of a theory on the origin of life which has been widely published elsewhere.  We included only those parts of the theory which are necessary for understanding our proposal concerning the specificity and we present the evidence supporting it. If we had included the long list of items, which according to the reviewer are missing from our paper, we would have produced a book, not a paper. The reviewer even criticizes us for not including a discussion of other theories on the origin of life! This is certainly an unfair criticism. It should, however, have been obvious to the reviewer that we have, in fact, published a book and many papers on the dissipative theory of the origin of life in which all of his/her questions have been addressed. The book and our papers are referenced throughout our manuscript. The proposal of our present paper on the origin of the early specificity between amino acids and the genome, however, is novel and is not found in our book or any previous paper.

3) As mentioned above, the reviewer insists that we should have presented more details concerning the dissipative theory for the origin of life but at the same time they ask us to remove all our figures which explain the main idea of the dissipative theory. Besides being in stark contradiction to the reviewer’s previous criticism, this would leave our paper completely incomprehensible.

4) Our main result is that it is those amino acids with characteristics which most foment photon dissipation and that could therefore most enhance the dissipation-replication relation, that are precisely those with the strongest stereochemical association with their anticodons or codons. Table 1 provides the evidence on which this result is based.  The section where we present the table is now called the “Results” , Section 6, and we have moved the table to this section rather than including it directly after the paragraph of its first mention (as instructed to do in Instructions for Authors). The reviewer appears to imply that we arbitrarily assigned characteristics to the amino acids; however, had the reviewer read section 5 they would have realized that we have always included a reference along with each assignment. Of course, all characteristics are valid for a range of pH values from slightly acid to slightly alkaline, which is the range for the Archean that has been discussed in the literature regarding the origin of life.

The reviewer suggests that our table is confusing and says that we assign both a strong and a weak sterochemical interaction to a particular amino acid. This should not be confusing as it is obvious from the table that these different assignments correspond to different codons which specify for the same amino acids. The different codons are explicitly listed in the table. It is difficult for us to imagine how there could be any confusion.

5) We have corrected the typos and an error in the figure caption that the reviewer has pointed out.

6) The reviewer suggests that we are presenting already published material as if it were new. This is not true as the editor can easily verify that wherever we include something published previously, we have always included a reference to that work. It seems that the reviewer has not paid attention to, and much less consulted, the extensive list of references that we have cited throughout our paper.

7) The reviewer makes a curious statement, “The universal evolution criterion of Glandsdorff and Prigogine is used by authors only as a window-dressing to impress readers without any invested effort to prove its relevance for photon-dissipation driven autocatalytic reactions presumably leading to nucleic acid enzyme-less replication and extension.”  

The paragraph concerning the universal evolution criterion was included in the first revision of our manuscript in direct response to a query from reviewer 2, not to “impress readers”, but rather to describe how one can expect an autocatalytic system to evolve through states of increasing entropy production based on the most general result obtained from classical irreversible thermodynamics which is known as the Glansdorff-Prigogine universal evolution criterion.  This criterion is valid for all systems falling within the domain of classical irreversible thermodynamics (that’s why it is called “universal”) and we argued that our system falls within this domain in the previous paragraphs. Even though our work is submitted to a journal whose audience is expected to have some knowledge of non-equilibrium thermodynamics, we understand how this paragraph might appear intimidating to someone with no or very little knowledge of non-equilibrium thermodynamics. We have re-written the paragraph in an endeavor to make it more transparent. We further emphasize to the reviewer that we had given a reference to an example in the book by Glansdorff and Prigogine for a chemical reaction and we now provide a further citation to a recent article of our own for a photochemical reaction (the production of Adenine from HCN) which presents the kinetics of the dissipative structuring exactly like what we described above.

The reviewer states “Line 95  - Non-linearity is not due (and not only due) to the feedback effect”.  Our statement at line 95 is “In inherently non-linear chemical or photochemical systems in which there is positive feedback…” Here it becomes evident the reviewer’s lack of knowledge non-equilibrium thermodynamic language since all chemical systems are inherently non-linear (whether there is feedback or not). Non-linear refers to the fact that the forces (affinities over the temperature, A/T) are not linearly proportional to the flows (rates of reaction) except very near to equilibrium. Therefore non-equilibrium chemical systems are inherently non-linear. The rest of our phrase “…in which there is positive feedback…” is in addition to the non-linearity inherent in chemical reactions and refers to those chemical reaction systems that are autocatalytic. Our phrase is correct and well written as stands; the confusion arises for the reviewer because of their lack of knowledge of non-equilibrium thermodynamic language.

We do not find any circularity of argument at line 140. Again, we believe that this misunderstanding derives from the reviewer’s lack of knowledge of non-equilibrium thermodynamics. If adenine, or any intermediate on the photochemical route to adenine, act as a catalyst for the production of adenine, then one can expect that the stationary state concentration of that intermediate or of adenine itself can become many orders of magnitude greater than what it would be if the situation was near to equilibrium. This is the proliferation of adenine, and this gives the relation between dissipation and replication. We cited a reference where this is shown for purely chemical reactions and now we include a new reference for photochemical reactions (in particular for adenine) at this point in the text in which we show this explicitly.

We changed “acid” to “acids” at line 171.

Reviewer states “Lines 307-318 and 434 – Confusing extension with replication”. We do not confuse extension with replication. In line 306-312 we talk about UVC-induced denaturing and in lines 312-315 we talk about overnight extension aided by Mg ions. Denaturing plus extension does give replication. In fact, on line 310 we state “Enzymeless photon-induced denaturing is only one half of the complete enzymeless replication problem, the other half being extension.”

Reviewer states “Lines 362-373 – Confusing and contradictory statements about which amino acid types were the most important for the origin of life: hydrophobic or amphipathic”. It is not just one characteristic (hydrophobic or amphiphatic) that makes an amino acid important for the origin of life, but as table 1 shows, those amino acids with the most characteristics important to photon dissipation are also those which seem to have been most important to the origin of life since these most strongly bind to their codon or anticodon.

Reviewer states “Lines 426-431- Would-would sequence about „thermodynamic selection“ proposal.” We don’t see what the reviewer is complaining about here, but we have improved the redaction of this sentence.

We thank the reviewer for their work although we strongly disagree with their general perception that our article is confusing or lacking logic.  This perception is in contrast to the perception of the other three reviewers, particularly reviewer 1 who suggested that “The authors have written an excellent paper …”, and found our paper “… well structured, well argued and clearly presented.” and “I am enthused by this novel idea of how the specificity of amino acids and their codons evolved.”  Reviewer 2 suggested that “…the text is well written…”, and reviewer 3 says that our “… point of view is justified” and “ … should be published…”..

We can understand how our article might appear confusing or intimidating to someone not from the area of non-equilibrium thermodynamics, who does not understand dissipative structuring, but our submission to Entropy is predicated on the assumption that many of our readers will be familiar with non-equilibrium thermodynamics and we have included sufficient references such that our proposal could be accessible to those without training, albeit by exerting some effort to become educated on the subject.

In the revised version, we have made extensive revisions to paragraphs (particularly those new paragraphs explaining the thermodynamics) to improve the understanding for those with little knowledge of non-equilibrium thermodynamics, we have renamed section 6 to “Results” and have moved the table to this section. Finally, we have corrected the typos and grammar errors pointed out by the reviewer and will re-submit the paper as a “hypothesis” paper.

Round 2

Reviewer 4 Report

It must be admitted that the efforts of this reviewer were not successful in the goal of improving the manuscript of Mejia & Michaelian so that it can be published in some quality journal such as Entropy. Except for some cosmetic changes, the revised manuscript is identical to the previous version. The authors did not observe the usual procedure of answering all the reviewer’s comments point by point. They ignored almost all of them, and they did not use the opportunity to address and correct several glaring weaknesses of their paper.

They renamed the section containing Table 1 as the Results section without introducing the most important of suggested corrections and explanations. This new “results” section does not provide any quantitative results. The only numbers (results) it contains are some cherry-picked reproductions of Yarus et al. results from 2009 that are utterly incomprehensible without Yarus et al. extensive explanations and descriptions.

Contrary to the suggestion of this reviewer, authors are persistent in their wish to publish again and again (for the third time in the case of Figure 1) the identical ill-rendered results (with overlapping textual contents inside the figure) they already published many times.

It did not bother the authors that this and other reproduced results from previous publications have no connection whatsoever with the present-day research accomplishments from the origin-of-life research. It is possible that this paper, now presented as the hypothesis paper, would make some sense 40 to 50 years ago, but not today.

I am not sure if other reviewers and the editors noticed that authors introduced the new pseudo-scientific concept of “microscopic dissipative structures” (MDS), which exists at thermodynamic equilibrium too. Apparently, for authors, the MDS is any structure that cannot be seen without a microscope, no matter how stable it is, and no matter how close it is to the thermodynamic equilibrium with its macroscopic environment. In particular, even the most durable of all biomacromolecules, the DNA, is considered by authors to be the MDS. Do authors know that DNA survived from long-extinct Neanderthals and Denisovans well enough to decode their genomes? It can also survive outside cells, for instance, in virions and phages, by not doing anything and certainly by not dissipating any energy forms. It is an extraordinary claim that nucleic acids are MDS. Authors are not even aware of how strange this claim is, and they do not invest any effort in defending it after being notified about this fact by this reviewer. It is hard to avoid the impression that authors lack the understanding of what dissipative structure is and what it is not. There is no question that dissipative structure subject to strong external forces can produce long-lasting changes in macroscopic and microscopic entities. Still, it is plain confusing to mix up the cause and consequence. It is just one among many other confusions perpetuated by authors in the revised manuscript, none adequately addressed by authors after my first review of their paper.